

# Characteristics of water jump for better performance in collegiate male 3000 m steeplechase

Yuya Maruo

Department of Physical Education, Tokyo Women's College of Physical Education, Kunitachi, Tokyo, Japan

## ABSTRACT

**Background**. The 3000 m steeplechase consists of 28 barriers and seven water-jumping obstacles. The water jump in the 3000 m steeplechase makes it different from the sprint hurdle events. It is important for coaches and athletes to understand how to clear the water jump successfully. I aimed to investigate whether the takeoff and landing distances for the water jump per lap differ between participants with good and worse records.

**Methods**. Data were collected from the men's 3000 m steeplechase races (heats) at Kanto Intercollegiate race. A total of 48 men's performances were analyzed (24 upper group, 24 lower group). Takeoff distance, landing distance and clearance time were analyzed. Takeoff distance, landing distance, total water jump distance and clearance time were subjected to mixed two-way ANOVAs with repeated factors of Lap (lap 1/lap 2/lap 3/lap 4/lap 5/lap 6/lap 7) with Group (upper group/lower group) as a between group factor.

**Results**. Takeoff distance was longer for upper group (1.43 m) than lower group (1.34 m) ($p = .01$). Landing distance was longer for upper group (2.95 m) than for lower group (2.74 m) ($p = .01$) and was longer for lap 1 (2.95 m) than last three laps (lap 5: 2.83 m, lap 6: 2.82 m, lap 7: 2.76 m) ($p = .01$).

**Discussion**. Individuals who were faster in 3000 m steeplechase exhibited longer water jump distance. The effect of fatigue might be greater for landing distance than for takeoff distance. Because the landing distance becomes shorter in the second half of the 3000 m steeplechase, it is important to note that athletes should aim to land as far away from the water pit as possible.

## INTRODUCTION

The 3000 m steeplechase consists of 28 barriers and seven water-jumping obstacles. Among all long-distance events, only the 3000 m steeplechase requires the skill to jump hurdles. There are differences in height of the obstacles between men (0.914 m) and women (0.762 m) races. When compared with other long-distance events, this race requires not only endurance but also power, technique (*Kipp, Taboga & Kram, 2017*), and pacing strategy (*Hanley & Williams, 2020*). It has been an official event since the 1983 World Championships and the women's race has been considered a world-record category since January 1, 2000. Only a few previous studies have investigated kinematical, physiological,

Corresponding author
Yuya Maruo, y-maruo@twcpe.ac.jp

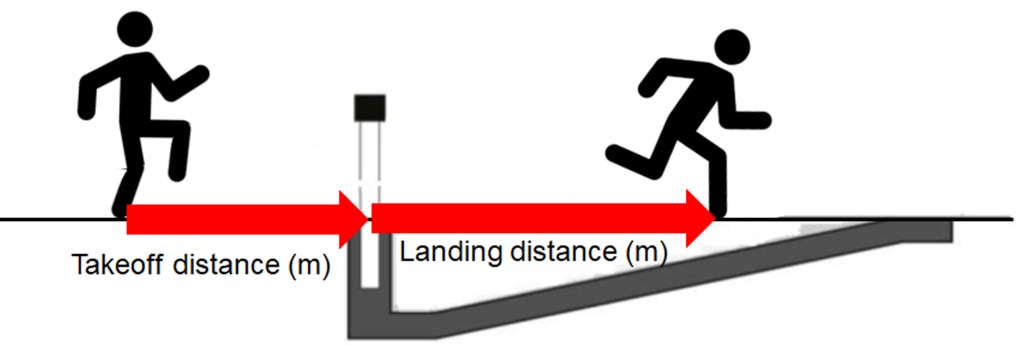

**Figure 1** Illustration for takeoff distance and landing distance in the water jump.

and psychological characteristics of the 3000 m steeplechase, mainly due to its lower popularity compared to other events, as well as its difficulty in biomechanical analysis (*Earl et al., 2015*; *Hanley, Bissas & Merlino, 2020*; *Hunter & Bushnell, 2006*; *Hunter, Lindsay & Andersen, 2008*; *Kipp, Taboga & Kram, 2017*).

The water jump in the 3000 m steeplechase makes it different from the other barriers in the 3000 m steeplechase (*Hunter, Lindsay & Andersen, 2008*; *Kipp, Taboga & Kram, 2017*). In the water jump, it is necessary to increase the running speed before the jump and the takeoff distance to ensure the athlete can make it onto the barrier. Takeoff distance refers to the horizontal distance from the takeoff toe and front edge of the barrier and landing distance refers to the horizontal distance from edge of the barrier and landing toe touching the water (*Hunter, Lindsay & Andersen, 2008*; Fig. 1). Previous studies suggested that water pit has a gradual upward slope, which makes landing distance a very important variable (*Hanley, Bissas & Merlino, 2020*; *Kipp, Taboga & Kram, 2017*) . If athletes land deeper in the water pit, it becomes difficult to exit. Therefore, a longer landing distance might be the key to success and better performance in the 3000 m steeplechase (*Hanley, Bissas & Merlino, 2020*). It is also important not to jump up in a vertical direction (*Hanley, Bissas & Merlino, 2020*; *Hunter, Lindsay & Andersen, 2008*). For instance, *Hunter, Lindsay & Andersen (2008)* found that the takeoff distance was 1.66 m for males and 1.41 m for females before the water jump. Women tended to step off closer to the obstacle than men because they are shorter. Although even-pace strategy and positive-pace strategy, which is slowing in the second half of the race, are considered desirable for long-distance running (*Hanon & Thomas, 2011*), it is difficult for athletes to use even-pace strategies due to the barriers and water jump in the 3000 m steeplechase. Investigating the ways to clear the water jump will be helpful for the race strategy and good performance.

Previous studies have investigated the gender differences (*Hanley, Bissas & Merlino, 2020*; *Hunter, Lindsay & Andersen, 2008*; *Hunter & Bushnell, 2006*; *Kipp, Taboga & Kram, 2017*) and variation in pacing (*Hanley & Williams, 2020*) for water jump in steeplechase. *Hunter, Lindsay & Andersen (2008)* investigated the characteristics of water jump techniques in steeplechase and found longer landing distances among men than among women. *Kipp, Taboga & Kram (2017)* investigated the ground reaction forces during water

jumps in steeplechase. These findings suggested that longer landing distances reflect better water jump technique. Previous findings suggest that the takeoff and landing distances per lap may vary depending on the race conditions, such as competitors and the category of competitions (*e.g.*, world-class competition, a track competition held at a certain university, (*Hanley, Bissas & Merlino, 2020*). Takeoff and landing distances for the water jump could be affected by fatigue and sprinting on the last lap. To the best of our knowledge, no studies have examined the variation per lap in the takeoff distance as well as in the landing distance for the water jump during a race.

This study aimed to investigate whether the takeoff/landing distances and clearance time for the water jump per lap differ between participants with good and worse time. Since athletes need to land as close to the end of the water-pit as possible (*Hunter, Lindsay & Andersen, 2008*), I focused on the distance for the water jump in the 3000 m steeplechase and compared the upper group with the lower group in terms of records. If the upper group preserve their energy in the second half of the race, they should be able to maintain nearly constant takeoff and landing distances over laps. In addition, the takeoff and landing distances should be longer for participants with better total time than for those with worse total time. I hypothesized that lower group would have shorter takeoff distances with each lap due to fatigue when compared with upper group. Therefore, the present study predicted the interaction between groups and laps. It is important for coaches and athletes to understand how to clear the water jump successfully.

## MATERIALS & METHODS

### Participants

Data were collected from the men's 3000 m steeplechase races (heats) at Kanto Intercollegiate race for division 1 and division 2 (Japan National Stadium, Tokyo). A total of 48 men's performances were analyzed (24 upper group; mean age $\pm$ SD = 20.8 $\pm$ 1.0 years, 24 lower group; mean age $\pm$ SD = 20.3 $\pm$ 1.3 years). Upper group consisted of the 24 participants who advanced to the final round. Lower group was selected from the bottom in the order of their time. Written informed consent was obtained from participants. This study was approved by the Ethics Committee of the Tokyo Women's College of Physical Education (Kenrinsin 2020-03).

### Procedure

The 3000 m steeplechase races were recorded by video camera (CASIO, EXILIM PRO EX-F1). The sampling rate was 300 Hz and the resolution was 512 $\times$ 384 px. The water jump was placed on the inside of the second bend. The camera was placed to film the athletes from a sagittal view at water jump on stadium. The camera was zoomed to include 5.8 m before and 11.3 m past the water jump.

### Data analysis

I compared upper group with lower group. The total time for the 3000 m steeplechase races were obtained from results documents (The Inter-University Athletic Union of Kanto, 2022). All jumps from participants were digitized using Kinovea (version 0.9.3; *Charmant,*

*2020*). Takeoff distance, landing distance and clearance time were analyzed. Before data collection, I measured 1.80 m before and 3.64 m past the water jump. At 1.80 m before water jump, there was a white marker for the javelin pit. These measurements were used to create a perspective grid using Kinovea, which made as a reference frame with dimensions of 3.66 m × 5.44 m. Measures of takeoff distance (the horizontal distance from the takeoff toe and front edge of the barrier) and landing distance (the horizontal distance from edge of the barrier and landing toe touching the water) were calculated using Kinovea. Endpoints of segments were determined by the researchers. Clearance time was the total time from takeoff before the barrier until the first contact made with the water.

Finishing time for running performance was subjected to unpaired *t*-test (upper group/lower group). Takeoff distance, landing distance, total water jump distance and clearance time were subjected to mixed two-way ANOVAs with repeated factors of Lap (lap 1/lap 2/lap 3/lap 4/lap 5/lap 6/lap 7) with Group (upper group/lower group) as a between group factor. Bonferroni correction was applied to *post-hoc* comparisons. All statistical analyses were conducted using JAPS (0.15.0.0).

## RESULTS

### Running performance

The average finishing time (min:s) were 8:54.81 ± SD 3.32 for upper group and 9:19.84 ± SD 10.15 for lower group. An unpaired *t*-test revealed that average finishing time was significantly shorter in upper group than in lower group ($t$ (46) = 11.57, $p = .01$, $d = 3.34$).

### Water jump distance

Table 1 shows each water jump distance for both groups. For takeoff distance, a two-way ANOVA revealed that the main effect for group was significant ($F$ (1, 46) = 8.57, $p = .01$, $\eta_p^2 = .08$). *Post-hoc* test revealed that takeoff distance was longer for upper group than lower group ($p = .01$). The ANOVA revealed no main effect for lap ($F$ (1, 46) = 0.54, $p = .74$, $\eta_p^2 = .01$). There was no interaction between lap and group ($F$ (6, 276) = 1.16, $p = .33$, $\eta_p^2 = .01$).

For landing distance, a two-way ANOVA revealed that the main effect for lap was significant ($F$ (6, 276) = 4.25, $p = .01$, $\eta_p^2 = .03$). *Post-hoc* test revealed that landing distance was longer for lap 1 than lap 5 ($p = .01$), lap 6 ($p = .01$), and lap 7 ($p = .01$). A two-way ANOVA also revealed that the main effect for group was significant ($F$ (1, 46) = 12.55, $p = .01$, $\eta_p^2 = .14$). *Post-hoc* test revealed that landing distance was longer for upper group than lower group ($p = .01$). There was no interaction between lap and group ($F$ (6, 276) = 0.79, $p = .55$, $\eta_p^2 = .01$).

For total water jump distance, a two-way ANOVA revealed that the main effect for group was significant ($F$ (1, 46) = 19.03, $p = .01$, $\eta_p^2 = .29$). *Post-hoc* test revealed that total water jump distance was longer for upper group than lower group ($p = .01$). A two-way ANOVA also revealed that the main effect for lap was significant ($F$ (6, 276) = 2.82, $p = .01$, $\eta_p^2 = .06$). *Post-hoc* test revealed that total water jump distance was longer for lap1 than lap7

**Table 1  Mean takeoff distance, landing distance and total water jump distance.** Takeoff distance (m), landing distance (m) and total water jump distance (m) for both groups (SD).

|  | Lap 1 | Lap 2 | Lap 3 | Lap 4 | Lap 5 | Lap 6 | Lap 7 |
|---|---|---|---|---|---|---|---|
| Takeoff distance for upper group | **1.43 (0.13)** | **1.43 (0.16)** | **1.40 (0.12)** | **1.39 (0.14)** | **1.43 (0.15)** | **1.43 (0.15)** | **1.45 (0.10)** |
| Takeoff distance for lower group | 1.33 (0.13) | 1.35 (0.20) | 1.36 (0.19) | 1.32 (0.17) | 1.35 (0.14) | 1.35 (0.14) | 1.30 (0.12) |
| Landing distance for upper group | **3.02 (0.29)** | **2.96 (0.19)** | **2.99 (0.22)** | **2.91 (0.27)** | **2.94 (0.23)** | **2.93 (0.29)** | **2.89 (0.28)** |
| Landing distance for lower group | 2.87 (0.26) | 2.75 (0.31) | 2.72 (0.37) | 2.75 (0.28) | 2.70 (0.31) | 2.70 (0.24) | 2.64 (0.24) |
| Total water jump distance for upper group | **4.45 (0.34)** | **4.40 (0.27)** | **4.40 (0.24)** | **4.31 (0.31)** | **4.38 (0.28)** | **4.38 (0.36)** | **4.35 (0.32)** |
| Total water jump distance for lower group | 4.21 (0.32) | 4.11 (0.36) | 4.09 (0.38) | 4.08 (0.39) | 4.06 (0.33) | 4.06 (0.27) | 3.94 (0.32) |

Notes.

The bold styling suggested that main effect for groups was significant ($p = .01$). Takeoff distance, landing distance and total water jump distance were longer for upper group than lower group ($p = .01$).

The underline suggested that the main effect for laps was significant ($p = .01$). Landing distance was longer for lap 1 than lap 5 ($p = .01$), lap 6 ($p = .01$), and lap 7 ($p = .01$). Total water jump distance was longer for lap 1 than lap 7 ($p = .01$).

**Table 2  Mean clearance time.** Clearance time (s) for both groups (SD).

|  | Lap 1 | Lap 2 | Lap 3 | Lap 4 | Lap 5 | Lap 6 | Lap 7 |
|---|---|---|---|---|---|---|---|
| Clearance time for upper group | 0.82 (0.07) | 0.80 (0.07) | 0.79 (0.07) | 0.78 (0.07) | 0.80 (0.06) | 0.81 (0.08) | 0.79 (0.06) |
| Clearance time for lower group | 0.77 (0.06) | **0.76 (0.06)** | 0.76 (0.08) | 0.77 (0.07) | 0.78 (0.06) | **0.80 (0.05)** | 0.79 (0.05) |

Notes.

The bold styling suggested that interaction between lap and group was significant ($p = .05$). Clearance time for lower group tended to be longer for lap 6 than lap 2 ($p = .06$).

The underline suggested that the main effect for lap was significant ($p = .01$). Clearance time was longer for lap 6 than lap 2 ($p = .03$), lap 3 ($p = .03$) and lap 4 ($p = .02$).

($p = .01$). There was no interaction between lap and group ($F (6, 276) = 0.71$, $p = .64$, $\eta_p^2 = .01$).

## Clearance time

Table 2 shows clearance time for both groups. For clearance time, a two-way ANOVA revealed that the interaction between lap and group was significant ($F (6, 276) = 2.11$, $p = .05$, $\eta_p^2 = .04$). *Post-hoc* test revealed clearance time for lower group tended to be longer for lap 6 than lap 2 ($p = .06$). A two-way ANOVA also revealed that the main effect for lap was significant ($F (2, 276) = 3.62$, $p = .01$, $\eta_p^2 = .02$). *Post-hoc* test revealed that clearance time was longer for lap 6 than lap 2 ($p = .03$), lap 3 ($p = .03$) and lap 4 ($p = .02$). There was no main effect for group ($F (1, 46) = 1.66$, $p = .20$, $\eta_p^2 = .04$).

## DISCUSSION

This study aimed to investigate whether the takeoff/landing distances and clearance time for the water jump per lap differ between participants with good and worse records. Performances of 28 male athletes were analyzed (24 upper group, 24 lower group) at the Kanto Intercollegiate race. The takeoff distance was longer in upper group than in lower group. The group effect on the takeoff distance was consistent with some of the

findings in previous studies. Athletes would need to approach the water jump from a farther distance for better timing in the 3000 m steeplechase. According to a previous study (*Hunter, Lindsay & Andersen, 2008*), the appropriate takeoff distance is 1.66 m for men. In the present study, the takeoff distance was 1.43 m for upper group during the race. On the one hand, the takeoff distance was 1.34 m for lower group during the race. Possibly, for Japanese university athletes, a longer takeoff distance leads to better performance in the 3000 m steeplechase. Since there was no variation in the takeoff distance during the race, it might not be susceptible to fatigue and pacing strategies. It is possible that the takeoff distance might not vary because a consistent technique is required to clear the water jump.

The landing distance was longer for upper group than for lower group, and it was longer for lap 1 than for the last three laps (laps 5, 6, and 7). These findings suggest that for better performance, athletes need to land closer to the end of the water pit. *Hunter, Lindsay & Andersen (2008)* suggested that landing distance was one of the important factors for better performance. In their study, the average landing distance was 2.85 m for men. In the present study, the landing distance for upper group was 2.89 m during the race. Our results were almost consistent with those from previous studies. Since landing deeper in the water pit makes it harder for the athletes to exit, landing closer to the end of the water pit would be the key to success and better performance in the 3000 m steeplechase. In addition, landing distance for both upper group and lower group got shorter with each lap in the second half of the race. The shorter landing distance in the second half of the race appears to reflect the effect of fatigue. To the best of our knowledge, this is the first demonstration of the effect of fatigue in a natural setting. Previous studies have reported mean values during races or experiments (*Hanley, Bissas & Merlino, 2020*; *Hunter & Bushnell, 2006*), and no studies have investigated the variation per lap in the takeoff and landing distances for the water jump during the race. The effect of fatigue was greater for the landing distance than for the takeoff distance. When athletes take off, there may be a distance at which it is easier to take off. Because the pushing motion is emphasized when jumping from a water jump (*Hanley, Bissas & Merlino, 2020*), so as the race progresses, it may become more difficult to push off the obstacles, resulting in a shorter landing distance.

Importantly, the main effects of group and lap independently influenced the landing distance. The landing distance differed according to the performance in the 3000 m steeplechase. According to previous studies on pacing for Olympic 3000 m steeplechase (*Hanley & Williams, 2020*), upper group increased their speed in the second half of the race. This result suggested that upper group for the Olympics were able to increase the pace in the second half of the race (especially in the last lap) with some extra energy. In the present study, supplementary focus only on the last lap revealed that the mean landing distance in lap 7 was 2.89 m for upper group and 2.64 m for lower group. Thus, it is possible that upper group saved their strength for the last lap. Longer landing distance may help to prevent the loss of energy and time in a water pit. This study focused on qualification for an intercollegiate race. Because it is unclear whether there are differences in jumping technique in the water jump depending on the level of competition and gender, further studies are needed to investigate the variation per lap for the water jump among athletes at various competitive levels.

I found the for clearance time interaction between lap and group, indicating that lower group tended to be longer for lap 6 than lap 2. A previous study suggested that average clearance time for men is 0.76 m during the race (*Hanley, Bissas & Merlino, 2020*). In this study, which conducted lap-by-lap analysis, it was revealed that for lower group the lap 6 had the longest clearance time at the water jump (0.80 m). In addition, the main effect for lap suggested that clearance time was longer for lap 6 than for lap 2, lap 3 and lap 4. Considering the results of the landing distance, lower group exhibited shorter jump distances and longer clearance times for the lap 6.

Faster athletes in the 3000 m steeplechase exhibited longer water-jump distances (takeoff, landing, and total water jump distances). In addition, throughout the race, the clearance time for the water jump remained relatively consistent. Superior performance for most of the long-distance events could be attributed to VO2max and running economy (*Conley & Krahenbuhl, 1980*; *Williams & Cavanagh, 1987*). However, *Earl et al. (2015)* found that performance in the 3000 m steeplechase was not related to the ratio of running economy, suggesting that better performance in the 3000 m steeplechase might be associated with VO2max and other factors such as strength, ability to change the pace, and jump technique. *Gabrielli et al. (2015)* suggested that other factors related to 3000 m steeplechase include the importance of muscle relaxation and posture, as well as techniques to conserve energy consumption and control breathing. For example, because Fartlek training is based on speed changes, it may be useful in learning the technique of jumping hurdles during long-distance running. I performed a simple analysis that focused exclusively on the jump distances of the water jump. The results revealed a statistically significant difference between the upper group and the lower group. Improving performance in the 3000 m steeplechase may be achieved by focusing on the simple technique of taking off from a farther distance and landing farther in the water pit during training.

This study has some limitations. Physiological changes such as heart rate during the 3000 m steeplechase were not recorded. Recording both physiological and kinesiological data would provide further evidence regarding the jump performance in the 3000 m steeplechase. In addition, the present study only examined the jumping distance and clearance time in the water jump technique. There are no data on the athletes' height and jump height in this study. To gain a more detailed understanding of this technique, it may be necessary to conduct further analyses of other factors such as running speed, athletes' height, jump height and joint angles.

## CONCLUSIONS

In conclusion, our results suggest that athletes with better finish time have longer takeoff and landing distances during the jump. Regardless of whether the finish time is better or worse, landing distance for the water jump gets shorter in the second half of the race. These findings may be affected by fatigue during the race. For athletes and coaches, it is advisable to be mindful of the need to takeoff from a distance and land at a distant water pit. Because the landing distance becomes shorter in the second half of the 3000 m steeplechase, it is important to note that athletes should aim to land as far away from the water pit as possible. This will make it easier for them to escape from the water pit.

## ACKNOWLEDGEMENTS

I would like to thank Editage for English language editing.

### Funding

The authors received no funding for this work.

### Competing Interests

The authors declare there are no competing interests.

### Author Contributions

- Yuya Maruo conceived and designed the experiments, performed the experiments, analyzed the data, prepared figures and/or tables, authored or reviewed drafts of the article, and approved the final draft.

### Human Ethics

The following information was supplied relating to ethical approvals (*i.e.*, approving body and any reference numbers):

The Ethics Committee of the Tokyo Women's College of Physical Education (Kenrinsin 2020-03)

### Data Availability

The raw measurements are available in the Supplementary Files.

### Supplemental Information

Supplemental information for this article can be found online at http://dx.doi.org/10.7717/peerj.15918#supplemental-information.

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
