# Peer review of "Characteristics of water jump for better performance in collegiate male 3000 m steeplechase"

_PeerJ, doi:10.7717/peerj.15918_

## Round 0.1 · original submission · Major Revisions

Dear Authors,

please make corrections according to the reviewers' remarks or write a detailed rPbuttal on a point-by-point basis.

Reviewer 1 ·

Basic reporting

No comment

Experimental design

No comment

Validity of the findings

No commment

Additional comments

Line 15 – although we normally write the unit with a space after the numeral (as you have done here), you don’t need to do this for this event as its name (as per World Athletics normal practice) is the 3000m steeplechase. You have omitted the space in the title, and you can also do so in the rest of your manuscript.
Line 17 – only a small point, but this seems to be a single-author study, so should this read “I” instead of “we”? Same comment for the rest of the paper where you use "we" or "our".
Line 31 – the word “pit” appears here on its own.
Line 38 – what are these heights? What do you mean by ”the height of the obstacle varies”? It is the same throughout the race.
Line 64 – I don’t know what you mean by the word “match”.
Line 78 – you need to explain better what you mean by qualifiers and non-qualifiers.
Line 85 – please report the SDs to one decimal place, as you have with the means.
Line 86 – what do you mean by “poor time”?
Line 99 – please report this to two decimal places (if it was 1.80 m, state this).
Line 115 – did you compare this performances statistically?
Results – you have written partial eta squared incorrectly. The 2 should be superscripted and the p subscripted.
Line 158 – note that Schmolinsky’s book is a guide on how to run rather than a series of experimental or observational studies. You would be better off comparing with results from published studies.
Line 188 – here is a good example of why using the term “qualifiers” is confusing in your study, as in the Hanley & Williams study it referred to those who did not make a global championship final – I don’t think this is what you mean in your study.
Line 210 – events between 1500m and 5000m tend to be VO2 max dominant, so this might be why running economy is not as important.
Line 232 – I think “worse” would fit better than “poor”.
References – these are not presented in a consistent manner, so please check these for consistency.
Table 1 – you seem to have presented the mean values in m, but the SDs in cm. Please make sure these are the same (as it currently is shown, the SDs are far bigger than the means).
Table 2 – please present the SDs to the same number of decimal places as the means.

Reviewer 2 ·

Basic reporting

The article examines the differences in takeoff and landing distances and clearance time for the water jump per lap during the 3000m steeplechase between high and low performing collegiate male athletes. The subject is interesting and relevant from the training and competition perspective, the English language is clear and professional and the structure of the article conforms to standards, yet there are some shortcomings that need to be addressed.

Experimental design

No comment.

Validity of the findings

No comment.

Additional comments

Remarks that should be addressed:
Line 29 – Abstract – “ps” – the letter “s” should be deleted
Line 31 – the word “pit.” at the end of the line should be deleted
Line 38 – Introduction - The mentioned differences in the standard height of obstacles should be shortly explained for the readers.
Line 42-45 – “Only a few previous studies…” – please, add the references of these studies at the end of this sentence.
Line 46 – It should be explained in more detail why the water jump is different from other barriers, and why this is important for the overall result. Explain that the water pit has a gradual upward slope, which makes landing distance a very important variable. Define here the terms take-off distance and landing distance (consider adding a photograph or illustration to that purpose).
Line 59 – “Hunter, Lindsay” – add “and Andersen” (the same in line 168)
Line 66-68 – the authors mention that “very few studies have examined…” – these studies should be referenced at this point and also in line 180
Line 199-201 – Discussion – “…it was revealed that the lap 6 had the longest clearance time” – take note this was true only for the non-qualifiers
Line 223-228 – Limitations – also mention that there are no data on the athletes’ height and jump height.
Table 1 – title – should it read: “Mean takeoff distance….”? I strongly suggest the authors add statistics to the table (significant differences)
Table 2 – title – should it also read: “Mean clearance time…”? As for the table 1, I strongly suggest the authors add statistics to the table (significant differences).

---

## Round 0.2 · Minor Revisions

Please revise according to the reviewers comments.

Reviewer 1 ·

Basic reporting

No comment

Experimental design

No comment

Validity of the findings

No comment

Additional comments

Thank you for making the recommended changes. Your paper reads much better now.

Reviewer 2 ·

Basic reporting

No comments.

Experimental design

No comments.

Validity of the findings

No comments.

Additional comments

The authors have addressed all the reviewers’ concerns and the quality of the manuscript is now improved.

Minor remark/suggestion:

Line 37-38 - "There are differences in height of the obstacles varies between men (0.914 m) and women (0.762 m) races." Delete the word varies and add the word races.

---

## Round 0.3 · accepted · Accept

Dear Authors,

Your manuscript is now acceptable for publication.